# Feed Regime Slightly Modifies the Bacterial but Not the Fungal Communities in the Intestinal Mucosal Microbiota of Cobia Fish (*Rachycentron canadum*)

**DOI:** 10.3390/microorganisms11092315

**Published:** 2023-09-14

**Authors:** Samira Reinoso, María Soledad Gutiérrez, Angélica Reyes-Jara, Magaly Toro, Katherine García, Guillermo Reyes, Wilfrido Argüello-Guevara, Milton Bohórquez-Cruz, Stanislaus Sonnenholzner, Paola Navarrete

**Affiliations:** 1Microbiology and Probiotics Laboratory, Institute of Nutrition and Food Technology (INTA), University of Chile, Avenida El Libano 5524, Macul, Santiago 7830490, Chile; soleguti@uchile.cl (M.S.G.); areyes@inta.uchile.cl (A.R.-J.); magaly.toro@inta.uchile.cl (M.T.); 2Centro Nacional de Acuicultura e Investigaciones Marinas, CENAIM, Escuela Superior Politécnica del Litoral, ESPOL Polytechnic University, Guayaquil 090211, Ecuador; guianrey@espol.edu.ec (G.R.); warguell@espol.edu.ec (W.A.-G.); mbohorq@espol.edu.ec (M.B.-C.); ssonnen@espol.edu.ec (S.S.); 3Millenium Institute Center for Genome Regulation (CRG), Santiago 8331150, Chile; 4Joint Institute for Food Safety and Applied Nutrition (JIFSAN), University of Maryland, College Park, MD 20910, USA; 5Instituto de Ciencias Biomédicas, Facultad de Ciencias de la Salud, Universidad Autónoma de Chile, Santiago 8900000, Chile; katherine.garcia@uautonoma.cl; 6Facultad de Ingeniería Marítima y Ciencias del Mar, FIMCM, Escuela Superior Politécnica del Litoral, ESPOL Polytechnic University, Guayaquil 090211, Ecuador

**Keywords:** cobia, *Rachycentron canadum*, microbiota, bacterial community, mycobiota, metabarcoding, functional analysis, 16S rRNA gene, ITS2

## Abstract

The bacterial community of the intestinal microbiota influences many host functions, and similar effects have been recently reported for the fungal community (mycobiota). Cobia is a tropical fish that has been studied for its potential in marine aquaculture. However, the study of its bacterial community has been underreported and the mycobiota has not been investigated. We analyzed the gut bacterial and fungal profile present in the intestinal mucosa of reared adult cobias fed two diets (frozen fish pieces (FFPs) and formulated feed (FF)) for 4 months by sequencing the 16S rRNA (V3-V4) and internal transcribed spacer-2 (ITS2) regions using Illumina NovaSeq 6000. No significant differences in the alpha diversity of the bacterial community were observed, which was dominated by the phyla Proteobacteria (~96%) and Firmicutes (~1%). Cobia fed FF showed higher abundance of 10 genera, mainly UCG-002 (Family *Oscillospiraceae*) and *Faecalibacterium*, compared to cobia fed FFPs, which showed higher abundance of 7 genera, mainly *Methylobacterium-Methylorubrum* and *Cutibacterium*. The inferred bacterial functions were related to metabolism, environmental information processing and cellular processes; and no differences were found between diets. In mycobiota, no differences were observed in the diversity and composition of cobia fed the two diets. The mycobiota was dominated by the phyla Ascomycota (~88%) and Basidiomycota (~11%). This is the first study to describe the gut bacterial and fungal communities in cobia reared under captive conditions and fed on different diets and to identify the genus *Ascobulus* as a new member of the core fish mycobiota.

## 1. Introduction

The intestinal microbiota is composed of a large and complex diversity of microorganisms including bacteria, yeasts, viruses, archaea and protozoa [1,2]. Microorganisms influence several host functions, including development, digestion, nutrition, disease resistance and immunity [3,4,5]. The autochthonous microbiota is considered of greater importance to the host because of its close association with the mucosal epithelium, stimulating the immune system, nutrient exchange and preventing colonization by pathogens [6]. 

There are at least 28,000 species of fish, representing almost half of all living vertebrates [7].They can live in marine, estuarine or freshwater habitats; they have different gut morphologies adapted to different dietary habits (herbivorous, omnivorous and carnivores) which can impact their gut microbiota composition [8]. Cobia (*Rachycentron canadum*) is a carnivorous marine benthopelagic fish that lives and feeds near the bottom, in mid-water or near the surface. It has a high potential for large-scale production in aquaculture facilities due to its rapid growth, easy adaptation to captivity, good meat quality and high demand from Asian consumers [9]. It is a tropical and subtropical species widely distributed worldwide except in the eastern Pacific [9]. A few studies have described the bacterial components of the cobia gut microbiota using 16S rRNA metabarcoding, revealing different compositions according to rearing conditions and fish development stages [10,11,12,13]. In these studies, cobia were reared in China, India and the South Caribbean Sea of Colombia. Ecuador is the only country in the eastern Pacific that cultivates cobia. This species has been introduced to diversify aquaculture production, and in spite of the role of the gut microbiota in fish health, no studies have been performed to characterize its composition.

The fungal community or mycobiota of fish has been poorly studied. We previously described the dominant gut mycobiota of marine fish including salmonids (*Salmo salar*, *Oncorhynchus kisutch* and *Oncorhynchus mykiss*), croaker (*Cilus gilberti*) and yellowtail amberjack (*Seriola lalandi*) using polymerase chain reaction and temporal temperature gradient gel electrophoresis (PCR-TTGE) [14]. We also recently identified the mycobiota of cultured red cusk-eel (*Genypterus chilensis*), palm ruff (*Seriolella violacea*) [15] and cobia [16]. Sequencing the ITS, especially the ITS2 region, located between the 5.8S and 28S rRNA genes, has been highly recommended to identify fungal microorganisms [17,18,19]. This approach has been applied mainly to describe the mycobiota of freshwater fish [20,21,22,23]. These studies showed that the fish gut mycobiota is highly dependent on the fish host, which justifies the analysis of each fish species.

Most of the microorganisms identified in fish mycobiota are yeasts belonging to the phyla Ascomycota and Basidiomycota, although filamentous fungi of Zygomycota [21], Cryptomycota, Neocallimastigomycota [22], and others have also been identified. Interestingly, some yeasts of this mycobiota have shown probiotic effects [15,16,24,25,26,27]. Considering the beneficial effect of some fungal commensals, further studies are needed to describe this community and understand how this mycobiota can respond to external factors such as diet. 

Providing healthy diets for a growing aquaculture is a challenging task, especially in fish with aquaculture potential such as cobia. Aquaculture centers often test different feed formulations for a specific fish species to improve its performance. However, studies in fish of advanced developmental stages (broodstock) are complex and the formulation of diets remains a challenge, in which case the use of fresh or frozen diets is an alternative. Although the effect of diet on the gut bacterial community of fish has been extensively studied [28], little is known about the effects of fresh/frozen and dry/formulated diets under similar culture conditions, considering that fresh/frozen diets are still widely used for reared marine fish in the reproductive phase. 

Considering that the gut mycobiota of marine fish has not been described, and neither has the microbiota of cobia reared in the eastern Pacific, we characterized the bacterial and fungal communities of the intestinal mucosa of cobia fish fed two diets (frozen fish pieces (FFPs) and a commercial formulated feed (FF)) by sequencing the 16S rRNA gene and ITS2 region. We also analyzed the effect of diets in the inferred bacterial functions using Tax4fun2 analysis.

## 2. Materials and Methods

The aim of this study was to describe the structure and composition of the microbiota of cobia under captive conditions and fed two types of diets. We determined the effect of diets on alpha diversity (primary outcome) of bacterial and fungal communities (mycobiota). As secondary outcomes, we determined the effect of diets on structure (beta diversity), differential abundance of microbial taxa (relative abundance of phyla/genera and LEfSe) and inferred functions. In addition, we described the composition of the core bacterial and fungal microbiota at the phylum and genus level shared by all fish and performed an exploratory analysis to evaluate the correlation between bacterial and fungal communities.

### 2.1. Fish and Experimental Design

This study was conducted in the National Center for Aquaculture and Marine Research of the ESPOL Polytechnic University (CENAIM–ESPOL), located in the province of Santa Elena, Ecuador (1°57′17.9″ S; 80°43′44.9″ W). Healthy cobia were obtained from a group of broodstock from CENAIM–ESPOL. Ninety fish were selected and randomly assigned to 4 tanks and reared under laboratory conditions in open flow systems at 35 g L^−1^ salinity, natural photoperiod (12:12 light dark), and 26.90 ± 0.72 °C. For four months, fish were fed ad libitum once a day with two types of diets; two tanks were fed formulated feed (FF) and two tanks were fed frozen fish pieces (FFPs) (diet composition is detailed in Appendix A). No antibiotics were used before or during the feeding period. A total of 18 healthy adult fish (9 fish per group) were randomly selected to analyze the gut microbiota (Appendix A). Fish health was assessed by visual inspection, i.e., those showing normal swimming and feeding behavior and no signs of bacterial, fungal or viral disease.

### 2.2. Intestinal Samples

For intestinal sampling (February 2021), fish were fasted 24 h prior to sacrifice by immersion in an overdose of anesthetic solution (50 mg L^−1^ eugenol, Eufar, Bogota, Colombia). The entire intestine of each fish was removed under sterile conditions and gently squeezed to remove its contents. The intestine was then dissected with sterile scissors and washed with sterile saline solution (NaCl 0.89%). To obtain intestinal mucosal samples, the intestinal epithelial layer was scraped with a sterile scalpel and carefully transferred to 1.5 mL tubes. These samples were immersed in 5 volumes of RNAlater (Invitrogen, Carlsbad, CA, USA) to preserve the DNA [29], and transported at room temperature to the Microbiology and Probiotics Laboratory of the Institute of Nutrition and Food Technology (INTA) in Santiago, Chile, for DNA extraction. 

### 2.3. DNA Extraction and Amplicon Sequencing 

To sequence the DNA regions of the bacterial and fungal communities, DNA was extracted from the intestinal mucosa of fish. Intestinal samples containing RNAlater were diluted 1:1 in 1X phosphate-buffered saline (PBS) (Thermo Fisher Scientific, Inc., Waltham, MA, USA) according to the manufacturer’s instructions [30]. They were centrifuged at 15,000× *g* and the supernatant was carefully removed. Total genomic DNA (gDNA) from pelleted samples was obtained using the DNeasy^®^ PowerSoil^®^ Kit (Qiagen, Hilden, Germany) according to the manufacturer’s instructions. A negative control consisting of DNA extraction from an equal volume of 1X PBS was included to control potential contamination of the reagents.

The gDNA from each sample was amplified to verify the presence of the 16S rRNA and ITS2 regions using primers 341F (5′-CCTACGGGAGGCAGCAG-3′) and 806R (5′-GGACTACHVGGGTWTCTAAT-3′), and ITS3F (5′-GCATCGATGAAGAACGCAGC-3′) and ITS4R (5′-TCCTCCGCTTATTGATATGC-3′), respectively. The amplification reaction was performed according to [16], with slight modifications.

The gDNA from samples with positive amplification using the above-mentioned primers was submitted to Novogene Corporation Inc. (Sacramento, CA, USA) for sequencing according to Novogene protocols. The bacterial community of the microbiota was identified using primers 341F and 785R (5′-GACTACHVGGGTATCTAATCC-3′) to sequence the V3–V4 region of the 16S rRNA gene. Mycobiota were identified by sequencing the ITS2 region, located between the 5.8S and 28S rRNA genes, according to recommendations [17,18,19], using primers ITS3F and ITS4R. PCR products of appropriate size were selected by 2% agarose gel electrophoresis. Quantified libraries were pooled and sequenced on an Illumina NovaSeq 6000 PE250 platform to generate 100,000 reads of 250 bp raw paired-end reads. At Novogene, paired-end reads were assigned to samples based on their unique barcode (demultiplex), then the barcode and primer sequence were truncated. Quality filtering was performed on the raw reads under specific filtering conditions (primers and adapters were trimmed from the reads; reads whose lengths were less than 60 bp after trimming the primers and adapters from the ends of the reads were removed; reads containing N > 10% were removed; and reads containing more than 50% bases with low quality (Qscore ≤ 5) were removed). 

### 2.4. Identification of Microbial Communities Using Bioinformatic Analysis

For the taxonomic identification of the reads obtained by sequencing, demultiplexed paired-end FASTQ reads and downstream analysis were processed in the DADA2 [31], phyloseq [32] and microbiome [33] packages in R statistical software version 4.2.2. The DADA2 package was used to clean and denoise the raw FASTQ reads, the Q-score was calculated, and trimming and filtering were performed to maintain high quality reads (Qscore ≥ 30), removing reads with unassigned nucleotides (NA). Then, the error rates of reads were calculated based on 1 × 10^8^ sequences; the reads were denoised and the amplicon sequence variants (ASVs) were inferred. The denoised reads were merged and chimeras were removed. Samples with low numbers of clean reads (<400) were excluded from the analysis. The ASVs of the bacterial community were assigned taxonomically using the SILVA database version 138.1 [34]; the UNITE ITS database version 9.0 [35] was used for the fungal community. A distance matrix was calculated using neighbor-joining to construct a phylogeny with the phyloseq package. Rarefaction was normalized to the data set containing the least number of sequences to remove heterogeneity between samples. For 16S rRNA analysis, the phyloseq object was filtered by removing the non-assigned (NA) phyla and genera. Only the unassigned phyla were removed for ITS analysis. 

### 2.5. Sample Size and Statistical Analysis

To estimate sample size, we focused on alpha diversity of the microbial communities (primary outcome). Based on previous studies describing the effect of diet on the fish gut microbiota and inter-individual variability [36,37], we estimated that 9 individuals per diet group would be needed to detect a between-group difference in a Chao1 index of 280, with a Cohen effect size of 1.5, with the assumptions of two-tailed, alpha value of 0.05, and a power of 0.80. The computation was performed by using GPower 3.1 software. In addition, we determined that this number of samples of each group was sufficient to describe most of the diversity (>90% of the identified ASVs), as previously described (Panteli 2020) (Appendix A).

For alpha diversity (primary outcome), the Chao1 index (based on the abundance of rare or infrequent species that have not been detected in the samples) was calculated. In addition, other alpha diversity indices were determined, such as richness (observed ASVs), Simpson (measures the probability that two randomly selected individuals belong to the same species), Shannon (measures the species diversity and equitability, i.e., how the relative abundances of different species are distributed in a community), and ACE (abundance-based coverage estimator, based on the species coverage in the sample). Alpha diversity was expressed as mean ± standard deviation, and t-tests were performed to detect significant differences (*p* ≤ 0.05) in those indices between diet groups after testing for normality (Shapiro–Wilks test [38]) and homogeneity of variances (Levene test).

To analyze the structure of the intestinal bacterial community of cobia fed the two diets, beta diversity was analyzed by principal coordinate analysis (PCoA) using unweighted Unifrac and weighted Unifrac (WUnifrac) distances. To identify significant differences (*p* ≤ 0.05), a PERMANOVA was performed. The Adonis2 and the Betadisper tests were performed to verify the homogeneity of variance between groups [39] using the vegan package. All diversity analyses were performed by the microbiome package. 

The relative abundances of the dominant (detection threshold > 5%) taxa (phyla and genera) per group were plotted. To determine the taxa contributing to the differences between cobia groups (differential abundance), a linear discriminant analysis (LDA) effect size analysis (LEfSe) was performed. This analysis was performed using the Microeco package with a significance level of *p* ≤ 0.05 and LDA threshold score ≥ 2.0 [40].

Bacterial functions were predicted using the ASVs against the pathway reference profiles (Ref99NR) from the Kyoto Encyclopedia of Genes and Genomes (KEGG) database. The Tax4fun2 package was used to predict the pathways, which were categorized into levels 1, 2 and 3. Similarities greater than 97% were considered orthologous KEGG groups (KOs) [41]. To identify significant differences (*p* ≤ 0.05) in predicted bacterial pathway composition between fish fed the two diets, PCoA and PERMANOVA were carried out. 

Bacterial and fungal taxa belonging to the core corresponded to those phyla and genera with prevalence > 70% and relative abundance > 0.1%. The microorganisms belonging to the core of each group were described using a heat map constructed by the microbiome package.

To determine the association between fungal and bacterial communities, a Spearman analysis was performed for correlation and plotted on a heatmap of the microbial abundances using the microbiome package. This plot shows taxa with a significant correlation (*p* ≤ 0.05) for phyla and genera.

R statistical software version 4.2.2 was used for all analyses.

## 3. Results

Eighteen healthy adults from the CENAIM–ESPOL center were sampled to identify the bacterial and fungal communities of the intestinal mucosa of cobias (*Rachycentron canadum*). Two fish were not analyzed because the amount of intestinal mucus was insufficient for DNA extraction. The bacterial community was identified in 15 samples: 7 fish fed with a commercially formulated feed (FF) and 8 fed with frozen fish pieces (FFPs). One sample (C16) from the FF diet group was excluded from the analysis due to the low number of clean reads (<400) after FASTQ processing. 

The mycobiota was identified in 12 samples, corresponding to 6 FF-fed and 6 FFPs-fed fish. Four samples were excluded from the analysis: two (C11 and C15) because they showed a low number of clean reads and two (C17 and C19) because they did not pass the quality control at the Novogene sequencing center (Appendix A). As expected, the negative control (contamination control for reagents) did not pass the DNA quality control and was not sequenced.

### 3.1. Bacterial Community Composition 

A total of 1,166,578 clean reads and 19,017 ASVs were obtained from 15 samples. Of the 19,017 ASVs, 18,343 (96.46%) were assigned to a phylum and 13,131 (69.05%) to a genus. The data were then rarefied to 52,690 reads, corresponding to sample C35, which contained the lowest number of reads (Appendix A). After rarefaction, we obtained a total of 790,350 reads and 9713 ASVs belonging to 24 phyla and 343 genera.

#### 3.1.1. Effect of Diet on Alpha Diversity (Primary Outcome)

No significant differences were observed in alpha diversity indices between the groups (Table 1). In general, we found a Chao1 index of 657.56 ± 138.59, richness of 647.53 ± 136.09, Simpson index of 147.44 ± 53.65, Inverse Simpson index of 0.99 ± 0.00, Shannon index of 5.32 ± 0.33 and ACE index of 655.66 ± 137.71.

#### 3.1.2. Effect of Diet on Beta Diversity

To analyze the structure of the intestinal bacterial community of cobia fed FF and FFP diets, beta diversity was analyzed by PCoA using unweighted Unifrac and weighted Unifrac (WUnifrac) distances (Figure 1). PCoA based on unweighted Unifrac showed significant differences (*p* = 0.001) between groups (Figure 1A), meaning that the taxa identified in the groups were phylogenetically different. In contrast, we did not observe differences according to diets (*p* = 0.189) using the WUnifrac distances (Figure 1B). This meant that the taxa identified in the groups were phylogenetically different, but they did not differ in their relative abundance. In particular, sample C15, corresponding to a fish fed FFPs, was grouped to the fish fed FF, showing that the relative abundance of its most abundant taxa was more similar to fish fed FF.

#### 3.1.3. Differential Abundances of Bacterial Communities

When we analyzed the relative abundance of bacterial communities, we observed that they were dominated by the phyla Proteobacteria (FF = 96.28 ± 1.27% and FFPs = 96.58 ± 1.95%), Firmicutes (FF = 1.32 ± 0.63% and FFPs = 1.23 ± 1.80%), Actinobacteriota (FF = 0.94 ± 0.52% and FFPs = 0.98 ± 0.44%) and Acidobacteriota (FF = 0.30 ± 0.11% and FFPs = 0.24 ± 0.17%) (Figure 2A). The bacterial community was dominated by the genera *Photobacterium* (FF = 86.74 ± 7.62% and FFPs = 70.70 ± 31.17%), *Vibrio* (FF = 4.55 ± 4.28% and FFPs = 20.30 ± 29.42%), *Catenococcus* (FF = 2.82 ± 2.84% and FFPs = 2.88 ± 1.42%) and *Enterovibrio* (FF = 0.40 ± 0.10% and FFPs = 0.90 ± 0.96%) (Figure 2B). The less abundant phyla and genera are shown in Appendix A. 

To identify the bacterial taxa that contributed to the differences of fish fed the two diets, we performed linear discriminant analysis effect size (LEfSe). Seventeen bacterial genera were differentially identified. Ten genera showed higher relative abundance in FF-fed fish: UCG-002 (family *Oscillospiraceae*), *Faecalibacterium*, NK4A136 group (family *Lachnospiraceae*), R-7 group (family *Christensenellaceae*), *Klebsiella*, UCG-005 (family *Oscillospiraceae*), *Parabacteroides*, *Desulfovibrio*, *Stenotrophomonas* and *Roseburia.* Seven genera showed higher relative abundance in FFPs-fed fish: *Actinobacillus*, *Marivita*, *Blastomonas*, *Pontimonas*, *Curvibacter*, *Cutibacterium* and *Methylobacterium-Methylorubrum* (Figure 3A). The abundance of all these genera were lower than 0.5% (Figure 3B). 

#### 3.1.4. Effect of Diet on Inferred Bacterial Functions

Finally, to detect differences in the inferred bacterial functions between the groups, we performed a Tax4fun2 analysis. A total of 21,620 KEGG genes were obtained. The relative abundances of bacterial pathways at three levels were similar between the bacterial communities of fish fed both diets (PERMANOVA, *p* ≥ 0.05). At level 1, we found metabolic functions (69.62 ± 1.12%), environmental information processing (12.64 ± 0.68%), cellular processes (8.70 ± 0.61%), genetic information processing (4.24 ± 0.24%), human diseases (3.29 ± 0.03%) and organismal systems (1.29 ± 0.03%) (Figure 4A). The most abundant pathways at level 2 were related to global and overview maps (34.81 ± 0.59%) and carbohydrate metabolism (10.33 ± 0.15%) (Figure 4B). The most abundant pathways at level 3 were related to metabolic pathways (13.40 ± 0.26%) and secondary metabolite biosynthesis (5.87 ± 0.17%) (Figure 4C).

#### 3.1.5. Core of the Bacterial Community

The core corresponded to those bacterial taxa showing a prevalence greater than 70% and a minimum detection threshold of 0.1% in all analyzed cobia. The bacterial core was composed of the phyla Proteobacteria, Actinobacteriota, Firmicutes and Acidobacteriota (Figure 5A), and the genera *Photobacterium*, *Vibrio*, *Catenococcus* and *Enterovibrio* (Figure 5B).

### 3.2. Fungal Community Composition

A total of 664,963 clean reads and 6036 ASVs were obtained, of which 1016 (16.83%) were correctly assigned to phylum and 396 (6.56%) to genus. Twelve samples were processed and rarefied to 1928 reads (Appendix A). After rarefaction, abundance and biodiversity were determined in 23,136 reads and 763 ASVs. Seven phyla and 85 genera were identified.

#### 3.2.1. Effect of Diet on Alpha Diversity (Primary Outcome)

Five alpha diversity indices were calculated and no significant differences between groups were found (Table 2): Chao1 index of 68.22 ± 22.46, Richness of 63.58 ± 19.08, Simpson index of 0.91 ± 0.08, Inverse Simpson of 15.98 ± 8.85, Shannon index of 3.15 ± 0.55 and ACE index of 67.60 ± 21.53.

#### 3.2.2. Effect of Diet on Beta Diversity

Beta diversity was used to analyze the structure of the intestinal fungal community of cobia fed FF and FFP diets. PCoA analysis using unweighted and weighted Unifrac distances showed no significant differences in the structure of the intestinal mycobiota of cobia fed with FF and FFPs (Appendix A). 

#### 3.2.3. Differential Abundances of Fungal Communities

The most abundant phyla identified in the intestinal mycobiota of cobia were Ascomycota (FF = 89.32 ± 5.43% and FFPs = 86.68 ± 9.67%) and Basidiomycota (FF = 10.26 ± 5.47% and FFPs = 12.60 ± 9.90%) (Figure 6A). Several ASVs were not assigned to a genus (FF = 41.98 ± 18.47% and FFPs = 59.12 ± 24.56%). Dominant genera were *Debaryomyces* (FF = 20.55 ± 28.65% and FFPs = 6.61 ± 7.91%), *Saccharomyces* (FF = 6.47 ± 7.58% and FFPs = 3.16 ± 2.23%), *Ascobolus* (FF = 7.29 ± 5.61% and FFPs = 1.75 ± 2.14%) and *Cladosporium* (FF = 1.23 ± 1.11% and FFPs = 2.76 ± 1.86%) (Figure 6B). No significant difference in the relative abundances of fungal taxa among phyla, classes or genera was detected in the mycobiota of fish fed the two diets. The less abundant phyla and genera are shown in Appendix A.

#### 3.2.4. Core of the Mycobiota

The core mycobiota was determined using a minimum detection threshold of 0.1% and a prevalence greater than 70% in all analyzed cobia. The core consisted of the phyla Ascomycota and Basidiomycota (Figure 7A), and the genera *Debaryomyces*, *Saccharomyces*, *Ascobolus* and *Cladosporium* (Figure 7B). 

### 3.3. Correlation between Bacterial and Fungal Communities

Finally, we explored some associations between fungal and bacterial taxa in all analyzed cobia. These results showed a significant positive correlation (*p* ≤ 0.05) between 54 fungal genera and at least one of the 96 bacterial genera (Appendix A). Although many genera were correlated, none belonged to the core microbiota of cobia gut. Correlations among phyla were not significant.

## 4. Discussion

Studies on the fish microbiota have increased in recent years due to its importance in host health, the advancement of sequencing technologies and the reduction in the analysis costs. However, considering the huge diversity in fish species, there is still not enough information about some fish species with aquaculture potential. In this study, the autochthonous bacterial and fungal communities present in the intestinal mucosa of cobia fed two diets (formulated feed (FF) or frozen fish pieces (FFPs)) were analyzed by sequencing the 16S rRNA gene and ITS2, respectively. To the best of our knowledge, this is the first study on the effect of diet on the microbiota of cobia reared in the eastern Pacific and the first to describe the mycobiota of a marine fish. 

### 4.1. Bacterial Community 

Diet is one of the most important factors influencing the diversity and composition of the bacterial community of the fish gut [42,43,44,45,46,47]. In this study, we compared the bacterial communities of the guts of fish fed formulated feed (FF) or frozen fish pieces (FFPs). 

We found no differences in alpha diversity between the groups using different diversity indices. The richness (647.53 ± 136.09) was similar to that found in other studies performed in juveniles (519 ± 17.24 [11], and 568.50 ± 151.44 [13]) and adults of cobia (506 [10]), and almost double that reported for larvae (318 ± 73.73 [48]). This is consistent with increasing diversity and richness as fish grow and develop [49]. The richness observed was lower compared to other herbivore fish such as the genus *Kyphosus* [50], and slightly higher compared to carnivorous fish such as yellowtail kingfish (*Seriola lalandi*) [51] and Atlantic salmon (*Salmo salar*) [52]. This result is similar to a previous report; lower gut bacterial diversity is generally observed in carnivores, and progressively increases in omnivores and herbivores [53]. 

In contrast, beta diversity analysis using unweighted Unifrac distances showed that the structure of the bacterial community was influenced by diet, meaning that the taxa identified in the two groups were phylogenetically different.We found that fish fed formulated feed were enriched in 10 genera; six of these belonged to the Firmicutes. In contrast, fish fed the FFPs diet were enriched in 7 genera; five of them belonged to the phylum Proteobacteria, abundant in marine wild fish [54,55]. Interestingly, several bacterial taxa that were enriched in formulated feed (*Oscillospiraceae*, *Faecalibacterium*, *Lachnospiraceae*, *Christensenellaceae*, *Parabacteroides* and *Roseburia*) produced short-chain fatty acids (SCFAs), especially butyrate, which have shown several beneficial effects on the host [56]. These SCFAs can be produced by the fermentation of dietary fiber. In our study, the formulated diet provides 3% dietary fiber, which could be fermented to SCFAs. Unfortunately, the content of dietary fibers of the frozen fish pieces was not determined, but dietary fiber did not occur naturally in this animal-based feed [57].

Little is known about the variation in the gut microbiota structure between fresh/frozen and dry/formulated diets, although fresh/frozen diets are still widely used for reared marine fish in the reproductive phase, where the nutritional requirements remain a challenge [58]. Some studies have compared the gut bacterial community of wild and farmed marine fish, i.e., live feed vs. formulated commercial feed, and found differences in the bacterial composition [54,55,59]; however other environmental factors may explain the observed differences. The FFPs (frozen fish pieces) diet used in this study consisted of fresh fish frozen at −20 °C for a maximum of 15 days, then cut, thawed, and fed as a diet. The freezing process was included to preserve the fish pieces, but we do not expect it to significantly alter their nutritional composition [60,61]. Freezing may inhibit the microbial spoilage and affect the viability of some microorganisms present in fresh fish pieces [61]; however, this analysis was not performed in this study. Future studies could address this issue and evaluate the effect of feed freezing on the gut microbiota of fish. 

The phylum Proteobacteria dominated the bacterial community in all cobia fish which was consistent with previous studies.Proteobacteria have been reported in larval [11,48], juvenile [10,11,13] and adult [12] healthy cobia, and other tropical marine aquaculture fish [62,63,64]. The most abundant genera were *Photobacterium*, *Vibrio*, *Catenococcus* and *Enterovibrio*, independently of the diet. Other studies have shown that the most abundant genera in the gut bacterial community of marine fish included *Vibrio* and *Photobacterium* [49]. The relative abundance of these genera has been associated with several benefits for the health of fish and other commercially important marine organisms [65,66]. It is important to note that some specific strains of these genera can act as pathogens, depending on their virulence repertoire and the health status of the fish. In particular, pathogens belonging to the genus *Photobacterium* have frequently been reported in cobia [67]. Some strains of the genus *Vibrio* are the causative agents of the disease called “vibriosis” [68]. Unfortunately, the sequencing of the 16S rRNA genes used in this study did not describe the pathogenic potential of the identified taxa, which is a known limitation of this technique.

The core bacterial taxa of cobia consisted of Proteobacteria, Actinobacteriota, Firmicutes and Acidobacteriota. The first three are the most abundant phyla identified in marine fish [49]. Although Acidobacteriota were not very abundant in the cobia bacterial communities, their prevalence was higher than 80% in the fish studied. Although this phylum is not often reported in a fish gut bacterial community, its relative abundance increased after supplementation of golden pompano (*Trachinotus ovatus*) with dihydromyricetin (DMY)-Zn (II) complex (50 mg kg feed^−1^). This additive is a bioactive flavonoid with beneficial effects in humans and animals, showing antioxidant, anti-inflammatory, anticancer, antidiabetic, antimicrobial and neuroprotective activities [69]. Most iron-reducing bacteria belong to the phyla Proteobacteria and Acidobacteriota, which may promote Zn utilization in the intestine [69]. According to this, the phylum Acidobacteriota may play an important role in cobia bacterial communities and fish health. It has been reported that abundance estimates based on 16S rRNA sequence counts tend to underestimate the abundance of taxa with low 16S rRNA copy numbers, such as the phylum Acidobacteriota in soil samples [70], which may explain the under-reporting of this phylum.

In this study, the abundances of bacterial functional pathways of the gut microbiota of cobia were not affected by diets. The most abundant pathways were associated with metabolism, environmental information processing and cellular processes. Although no functional studies have been reported in cobia, similar functions have been inferred from the gut bacterial communities of other marine fish [71,72]. 

### 4.2. Fungal Community

The mycobiota of the fish gut has been less studied compared to the bacterial community. Few fish mycobiota have been characterized using next-generation sequencing (NGS), and they have only been studied in freshwater fish [20,21,22,23,73]. The mycobiota of some marine fish have been studied using PCR-TTGE (temporal temperature gradient gel electrophoresis) [14].

In this study, the mycobiota was less affected by the diet compared to the bacterial community. This suggests that both microbial communities, despite sharing the same intestinal mucosal habitat, may not share the same ecological niche. The mycobiota may depend mostly on the nutrients present in the intestinal mucus, which has been shown to be used by fungal gut microorganisms [74]. In addition, previous studies describing the alpha diversity of the mucosal (autochthonous) microbiota (fungal and bacterial communities) is consistent with previous studies indicating that this microbiota tends to be highly conserved in response to dietary changes compared to the allochthonous microbiota [75].

Diets did not affect the composition of the mycobiota in our study; however, it is necessary to note that a high percentage of ASVs were not assigned taxonomically, which could explain this result. Studies based on NGS, like this study, usually identify Ascomycota as the dominant phylum in the gut mycobiota of fish, as has been reported [20,21,22,73]. Ascomycota and Basidiomycota have also been reported as the most abundant phyla in the aquatic environment [76], which may explain their presence in the fish gut. It should be noted that the presence of some opportunistic pathogenic fungi has been reported in fish [77]. Fungal infections in fish are generally considered secondary to some other factor or pathogen, a consequence of water quality problems, poor condition, trauma (rough handling or aggression), bacterial diseases or parasites [78]. To date, they have not been reported in cobia. Some opportunistic fungi belong to the genera *Saprolegnia*, *Branchiomyces*, *Achly* and *Aphanomyces* of Phylum Oomycota [78]. 

*Debaryomyces*, *Saccharomyces*, *Ascobolus* and *Cladosporium* were the most abundant genera of fungi. It should be noted that a large part of the reads could not be assigned to a genus, as has been previously reported [20,21,23]. This could be due to the lack of studies on mycobiota and the limited availability of databases [19]. Also, the number of sequences in fungal databases is lower compared to bacterial databases [34,35]. 

*Debaryomyces* was the most abundant genus identified in this study. *Debaryomyces* are ubiquitous in marine and other aquatic environments [79]. They are abundant in the mycobiota of rohu (*Labeo rohita*) [22] and zebrafish (*D. rerio*) [21], and they have also been identified as part of the core of several marine fish [14]. Several strains have been isolated from fish intestines using culture methods, and some have been proposed as potential probiotics for fish [15,16,24,25]. *Saccharomyces* is another genus frequently reported in aquatic environments and has also been evaluated for its probiotic potential [80,81,82]. *Ascobolus* is a coprophilous genus [83] not previously reported in fish. This genus usually lives on dung or rotting plant remains and has a worldwide distribution [84]. It has been characterized as having asci with an operculum [85], whose ascospores pass intact through animal alimentary canals and are often stimulated to germinate by this passage [84]. Further studies are needed to explore the role of this genus in fish health. 

We found that the cobia’s core mycobiota was dominated by *Debaryomyces*, *Saccharomyces*, *Ascobolus* and *Cladosporium*. As reported in other studies, *Debaryomyces* and *Saccharomyces* are members of the abundant mycobiota of fish [14,21]. *Ascobolus* showed low abundance (4.52 ± 4.97%) but was present in 91.67% of the samples, with a prevalence similar to that of *Cladosporium*, which had an abundance of 1.99 ± 1.66%. In contrast to *Ascobolus*, *Cladosporium* has previously been reported in fish such as tilapia (*O. mossambicus*) and bighead carp (*A. nobilis*) [23], but not as part of the core mycobiota. More studies are needed to disentangle the role of these fungi in fish health.

### 4.3. Correlation between Bacterial and Fungal Communities

Studies in humans and mice in recent decades have explored the specific cross-kingdom interactions between fungi and bacteria, showing that this relationship can be antagonistic, pathogenic or protective [86]. Studies in fish have shown a protective effect of some yeast isolates against bacterial pathogens, as described above. However, this was the first study exploring the association between the whole fungal and bacterial communities within the intestinal mucosa of fish using next-generation sequencing. We have detected many significant associations between genera, although these taxa do not belong to the core microbiota. Consequently, future studies are needed to disentangle the role of these interactions in fish intestinal homeostasis and health. 

## 5. Conclusions

No significant differences were observed in the alpha diversity of the autochthonous bacterial and fungal communities present in the intestinal mucosa of cobia fed two different diets. However, some differences in the structure of the bacterial community were detected between diets. No differences were detected in the inferred functional capacity of the bacterial community. Interestingly, we identified *Ascobolus*, which has never been reported in fish guts. This study may help us to define a baseline for new research in cobia. Considering that our microbial identification and function inference were based on high-throughput DNA sequencing, further studies are needed to confirm these results with qPCR or culture methods to isolate these microorganisms and explore their functions and effects on fish health. 

## Figures and Tables

**Figure 1 microorganisms-11-02315-f001:**
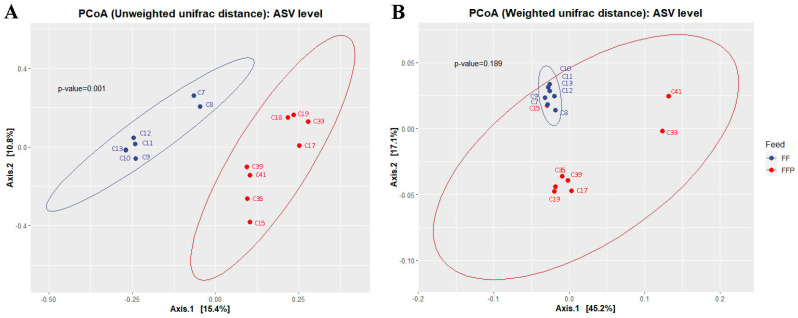
PCoA analysis of bacterial communities of cobia (*Rachycentron canadum*) fed formulated feed (FF, small blue dots) and frozen fish pieces (FFP, small red dots) using ASVs. (**A**) Unweighted Unifrac distance, significant differences between groups; the principal components explained 15.4 and 10.8% of data variance; and (**B**) weighted Unifrac distance, non-significant differences between groups; the principal components explained 45.2 and 17.1% of data variance. The ellipses (blue and red) represent 95% confidence level for each group for a multivariate normal distribution.

**Figure 2 microorganisms-11-02315-f002:**
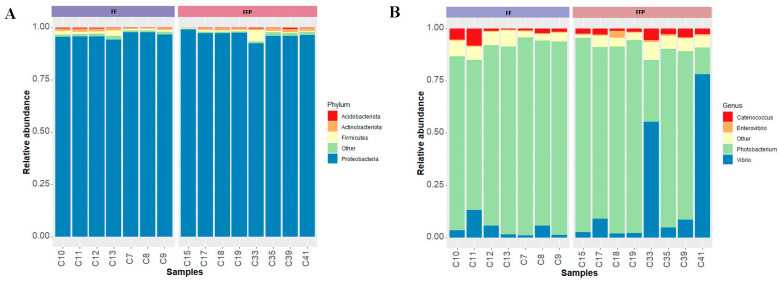
Dominant bacterial communities identified in the intestinal mucosa of cobia (*Rachycentron canadum*) fed formulated feed (FF) and frozen fish pieces (FFP). (**A**) Bacterial phyla and (**B**) bacterial genera composition. Bacterial taxa in this figure had a prevalence and detection threshold greater than 50% and 1%, respectively.

**Figure 3 microorganisms-11-02315-f003:**
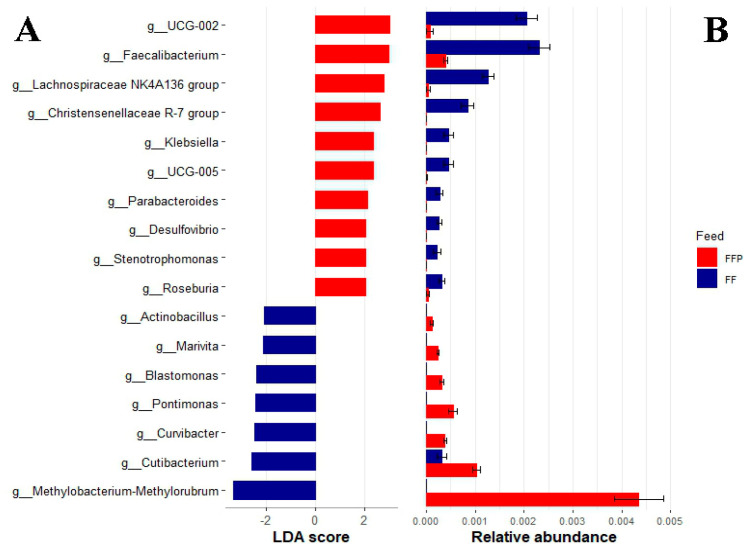
(**A**) Linear discriminant analysis (LDA) effect size (LEfSe) performed on the bacterial communities at the genus level for cobia (*Rachycentron canadum*) fed FF (formulated feed) and FFP (frozen fish pieces); and (**B**) relative abundance of each genus in each fish group (on the *x*-axis abundance ranges from 0 to 1).

**Figure 4 microorganisms-11-02315-f004:**
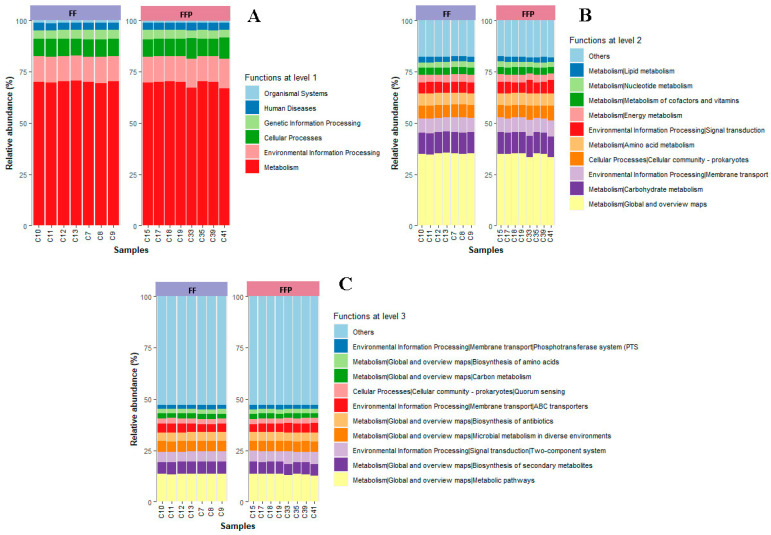
Relative abundance of bacterial pathways of cobia (*Rachycentron canadum*) fed with FF, formulated feed, and FFP, frozen fish pieces. (**A**) At level 1, (**B**) at level 2 and (**C**) at level 3.

**Figure 5 microorganisms-11-02315-f005:**
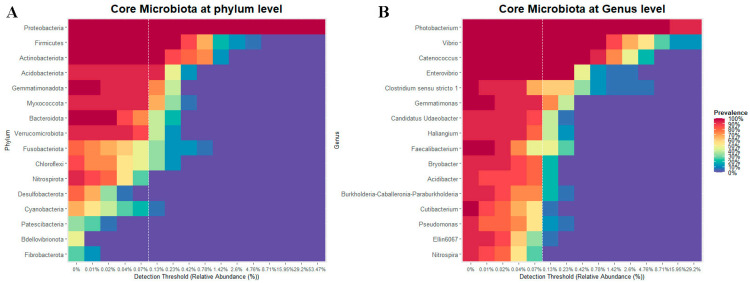
The core bacterial communities of cobia (*Rachycentron canadum*) fed formulated feed (FF) and frozen fish pieces (FFP). (**A**) At the phylum level and (**B**) at the genus level. The core included those taxa that showed a relative abundance > 0.1% (dashed white line) and a prevalence > 70%.

**Figure 6 microorganisms-11-02315-f006:**
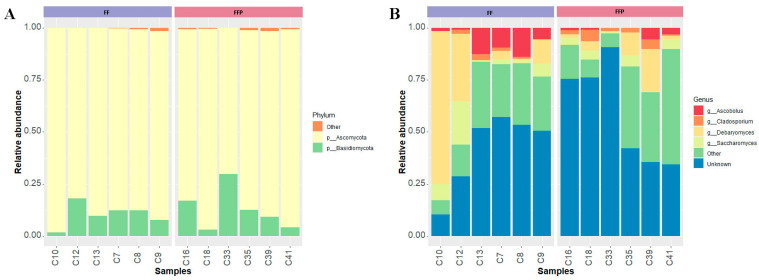
Dominant mycobiota in the intestinal mucosa of cobia (*Rachycentron canadum*) fed formulated feed (FF) or frozen fish pieces (FFP). (**A**) Relative abundance at the phylum level and (**B**) relative abundance at the genus level. Fungal taxa in this figure had a prevalence and detection threshold greater than 50% and 1%, respectively.

**Figure 7 microorganisms-11-02315-f007:**
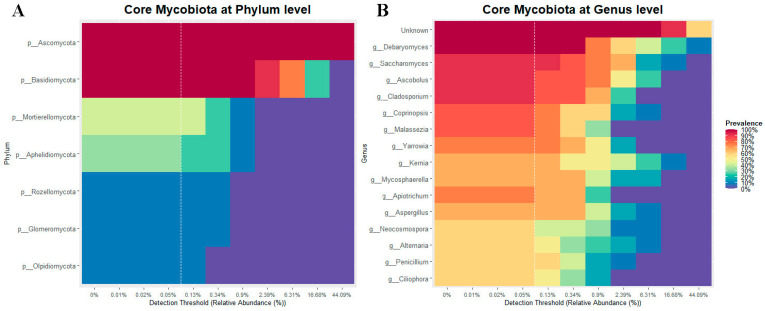
The core mycobiota of cobia (*Rachycentron canadum*) fed formulated feed (FF) and frozen fish pieces (FFP). (**A**) At the phylum level and (**B**) at the genus level. The core included those taxa that showed a relative abundance > 0.1% (dashed white line) and a prevalence > 70%.

**Table 1 microorganisms-11-02315-t001:** Alpha diversity indices of cobia intestinal bacterial communities. Diet: FF, formulated feed; and FFP, frozen fish pieces. Mean ± standard deviation.

Index	Diet		*p*-Value
	FF	FFP	
Chao1	645.22 ± 126.38	668.37 ± 156.33	0.760
Richness	634.00 ± 125.32	659.38 ± 152.42	0.733
Simpson	0.99 ± 0.00	0.99 ± 0.00	0.127
Inv. Simpson	124.96 ± 36.71	167.12 ± 60.44	0.133
Shannon	5.18 ± 0.31	5.45 ± 0.30	0.112
ACE	642.51 ± 125.57	667.17 ± 155.20	0.743

**Table 2 microorganisms-11-02315-t002:** Alpha diversity indices of cobia intestinal fungal communities. Diets: FF, formulated feed; and FFP, frozen fish pieces. Mean ± standard deviation.

Index	Diet		*p*-Value
	FF	FFP	
Chao1	68.29 ± 30.50	68.14 ± 13.39	0.148
Richness	62.33 ± 25.70	64.83 ± 11.69	0.109
Simpson	0.89 ± 0.11	0.92 ± 0.03	0.872
Inv. Simpson	16.55 ± 10.20	15.42 ± 8.22	0.336
Shannon	3.07 ± 0.75	3.22 ± 0.31	1.000
ACE	67.12 ± 28.92	68.09 ± 13.51	0.109

## Data Availability

All the sequence datasets are available on the NCBI Sequence Read Archive under the SRA study PRJNA925082 (accession numbers SRX19062117-SRX19062136).

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
