# Peer review of "Feed Regime Slightly Modifies the Bacterial but Not the Fungal Communities in the Intestinal Mucosal Microbiota of Cobia Fish (*Rachycentron canadum*)"

_microorganisms, 2023, doi:10.3390/microorganisms11092315_

Round 1
Reviewer 1 Report
In my opinion, this article presents the following problems.
There is no a clear experimental design and the design that appears to have been adopted may not be optimal, also given the low number of subjects, that, moreover, can hardly support generalization of results ; the randomization procedure for the definition of the experimental groups is not specified. The background of bioinformatic and statistical procedures used to analyze the data is not defined and explained with sufficient clarity regarding their meaning and effectiveness.
Some abbreviations (for example "PCoA") are not explained.
I would suggest to the authors to rewrite the article in the experimental part which concerns the setting of the study and its analysis by clearly defining, keeping the following order, the experimental design, the randomization procedure, the hypotheses to be tested statistically, the outcomes to be analyzed and, for each hypothesis, the statistical procedures to be used, the meaning of each of which must be explained and its use justified on the basis of the literature and the statistical hypothesis to be tested; only as the final item, the specific procedure of the adopted software should be given. Also the results should be formatted according the suggestions given above.
Author Response
We thank the reviewer for taking the time to read and review our manuscript. We agree with all the reviewer’s comments and incorporate all his suggestions in this new version. Please find below the answers to each comment.
Comments and Suggestions for Authors
Reviewer 1: In my opinion, this article presents the following problems.
There is no a clear experimental design and the design that appears to have been adopted may not be optimal, also given the low number of subjects, that, moreover, can hardly support generalization of results; the randomization procedure for the definition of the experimental groups is not specified. The background of bioinformatic and statistical procedures used to analyze the data is not defined and explained with sufficient clarity regarding their meaning and effectiveness.
I would suggest to the authors to rewrite the article in the experimental part which concerns the setting of the study and its analysis by clearly defining, keeping the following order, the experimental design, the randomization procedure, the hypotheses to be tested statistically, the outcomes to be analyzed and, for each hypothesis, the statistical procedures to be used, the meaning of each of which must be explained and its use justified on the basis of the literature and the statistical hypothesis to be tested; only as the final item, the specific procedure of the adopted software should be given. Also, the results should be formatted according to the suggestions given above.
Response: We fully agree with the reviewer and in this new version we have rewritten the Materials and Methods section according to the recommended structure. We have also presented the results in the same order. In addition, we have included in the M&M section the calculation of the sample size according to the primary outcome. Furthermore, we determined that the number of samples used was sufficient to describe most of the diversity (>90%) of each treatment, as recommended by (Panteli et al., 2020). This analysis was performed for both bacterial and fungal communities and has been included as a Supplementary Figure 5.
References:
Panteli, N., Mastoraki, M., Nikouli, E., Lazarina, M., Antonopoulou, E., & Kormas, K. A. (2020). Imprinting statistically sound conclusions for gut microbiota in comparative animal studies: A case study with diet and teleost fishes. Comparative Biochemistry and Physiology Part D: Genomics and Proteomics, 36, 100738. https://doi.org/10.1016/J.CBD.2020.100738
Reviewer 1: Some abbreviations (for example "PcoA") are not explained.
Response: We agree with the reviewer. We have included the meaning of the abbreviation in line 209, where it is mentioned for the first time.

Reviewer 2 Report
In this study, the authors analyzed the gut microbial profile present in the intestinal mucosa of reared adult cobias fed two diets by sequencing the 16S rRNA (V3-V4) and Internal Transcribed Space-2 (ITS2) regions. This work has been useful in the culture of Cobia fish, and the authors present their data quite well. Based on this aspect, this manuscript has certain scientific value, but there are some defects in the experimental methods and conclusions, and it lacks specific descriptions of some information.
1. I didn't find an animal welfare approval number, which is important.
2. The conclusions in this paper are all based on the sequencing results of 16S RNA, including KEGG analysis and so on. I think the authors should have selected at least some genes for qPCR validation, and the current conclusions are not very convincing.
3. In Figure 1A, one of the samples in the FFP group is clustered together with the FF group (Maybe C15?). P-value=0.189. This does not indicate a significant difference between the two groups.
4. The resolution of Figure 1 is too low to see the clustering of each sample.
5. C15 presents quite a different cluster between weighted and unweighted unifrac analysis, The author should explain and discuss it. It is not enough to mention it in line 218.
6. The discussion is well-written, but seems to lack an analysis of the causes of the differences brought about by "frozen".
7. I think the discussion part of this paper can be reduced appropriately to make the whole text more concise.
8. In Figure 5 and Figure 6, it is suggested to use different colours for FF and FFP, currently both have grey backgrounds and are not easily distinguishable.
Author Response
We thank the reviewer for taking the time to read and review our manuscript. We agree with all the reviewer’s comments and incorporate all his suggestions in this new version. Please find below the answers to each comment.
Comments and Suggestions for Authors
Reviewer 2: In this study, the authors analyzed the gut microbial profile present in the intestinal mucosa of reared adult cobias fed two diets by sequencing the 16S rRNA (V3-V4) and Internal Transcribed Space-2 (ITS2) regions. This work has been useful in the culture of Cobia fish, and the authors present their data quite well. Based on this aspect, this manuscript has certain scientific value, but there are some defects in the experimental methods and conclusions, and it lacks specific descriptions of some information.
I didn't find an animal welfare approval number, which is important.
Response: We thank the reviewer for this comments. We have included in line 612-617 the animal welfare approval number.
Line 574-579: “Institutional Review Board Statement: The fish species (Rachycentron canadum) used in this study is included in the list of species authorized for aquaculture activities in Ecuador (MAGAP-INP-2015-0606-M and MAP-SUBACUA-2017-5899-M), and the use of this fish species for research under animal welfare condition at the National Center for Aquaculture and Marine Research CENAIM-ESPOL, has been authorized by the Ministry of Environment of Ecuador (MAAE-DZ5-2021-6368-O)”.
The first document refers to the list of species authorized for mariculture activities in Ecuador, and the second is the authorization to conduct research on these species within the CENAIM-ESPOL Research Center, in compliance with animal welfare guidelines.
Reviewer 2: The conclusions in this paper are all based on the sequencing results of 16S RNA, including KEGG analysis and so on. I think the authors should have selected at least some genes for qPCR validation, and the current conclusions are not very convincing.
Response: We agree with the reviewer's comment, and we have rewritten the conclusions to consider this important point.
Line 530-539: No significant differences were observed in the alpha diversity of the autochthonous bacterial and fungal communities present in the intestinal mucosa of cobia fed two different diets. However, some differences in the structure of the bacterial community were detected between diets. No differences were detected in the inferred functional capacity of the bacterial community. Interestingly, we identified Ascobolus, which has never been reported in fish guts. This study may help us to define a baseline for new research in cobia. Considering that our microbial identification and function inference were based on high throughput DNA sequencing, further studies are needed to confirm these results with qPCR or culture method to isolate these microorganisms and explore their functions and effects on fish health.
Reviewer 2: In Figure 1A, one of the samples in the FFP group is clustered together with the FF group (Maybe C15?). P-value=0.189. This does not indicate a significant difference between the two groups.
We thank the reviewer for this comment. There is indeed an error in the names of the figures, which we have already corrected it in the manuscript.
Line 265-268: “PCoA based on unweighted Unifrac showed significant differences (p = 0.001) between groups (Figure 1A), meaning that the taxa identified in the groups are phylogenetically different. In contrast, we did not observed differences according to diets (p = 0.189) using the WUnifrac distances (Figure 1B)”.
Reviewer 2: The resolution of Figure 1 is too low to see the clustering of each sample.
Thank you for your comment. In this new version we have improved the resolution of this figure.
Reviewer 2: C15 presents quite a different cluster between weighted and unweighted unifrac analysis. The author should explain and discuss it. It is not enough to mention it in line 218.
We agree with the reviewer. We have added an explanation regarding this observation.
Line 268-272: “This means that the taxa identified in the groups are phylogenetically different, but they did not differ in their relative abundance. In particular, sample C15, corresponding to a fish fed FFP, was grouped to the fish fed FF showing that the relative abundance of its most abundant taxa was more similar to fish fed FF”.
Reviewer 2: The discussion is well-written but seems to lack an analysis of the causes of the differences brought about by "frozen".
We thank the reviewer for this interesting point. We have discussed this issue in this new version of the manuscript:
Line 427-433: “The FFP (frozen fish pieces) diet used in this study consisted of fresh fish frozen at -20 °C for a maximum of 15 days, then cut, thawed, and fed as a diet. The freezing process was included to preserve the fish pieces, but we do not expect it to significantly alter their nutritional composition [1,2]. Freezing may inhibit the microbial spoilage and affect the viability of some microorganisms present in fresh fish pieces [2]; however, this analysis was not performed in this study. Future studies could address this issue and evaluate the effect of feed freezing on the gut microbiota of fish”.
References:
- Nazemroaya, S.; Sahari, M.A.; Rezaei, M. Effect of Frozen Storage on Fatty Acid Composition and Changes in Lipid Content of Scomberomorus Commersoni and Carcharhinus Dussumieri. J. Appl. Ichthyol. 2009, 25, 91–95, doi:10.1111/J.1439-0426.2008.01176.X.
- Nakazawa, N.; Okazaki, E. Recent Research on Factors Influencing the Quality of Frozen Seafood. Fish. Sci. 2020, 86, 231–244, doi:10.1007/S12562-020-01402-8/FIGURES/11.
Reviewer 2: I think the discussion part of this paper can be reduced appropriately to make the whole text more concise.
We agree with the reviewer. We have reduced some section of the Discussion.
Reviewer 2: In Figure 5 and Figure 6, it is suggested to use different colours for FF and FFP, currently both have grey backgrounds and are not easily distinguishable.
Thank you for your comment, we have changed the colors of the figures 2, 5, and 6 to make them clearer.
